# Vaccination: Adherence and Hesitancy among Pregnant Women for COVID-19, Pertussis, and Influenza Vaccines

**DOI:** 10.3390/vaccines12040427

**Published:** 2024-04-17

**Authors:** Gabriele Filip, Alessia Sala, Veronica Modolo, Luca Arnoldo, Laura Brunelli, Lorenza Driul

**Affiliations:** 1Department of Obstetrics and Gynaecology, ASUFC, Ospedale Santa Maria Della Misericordia, 33100 Udine, Italy; lorenza.driul@asufc.sanita.fvg.it; 2Department of Medical Area, University of Udine, 33100 Udine, Italy; sala.alessia@spes.uniud.it (A.S.); veronica.modolo1@gmail.com (V.M.); luca.arnoldo@asufc.sanita.fvg.it (L.A.); laura.brunelli@asufc.sanita.fvg.it (L.B.); 3Accreditation, Quality and Clinical Risk Unit, Friuli Centrale Healthcare University Trust, 33100 Udine, Italy

**Keywords:** pregnancy, vaccination, prevention, hesitancy, adherence

## Abstract

In the realm of antenatal care, vaccinations serve as a cornerstone, crucial for safeguarding the health of both the mother and the fetus, while also extending protection to the newborn against communicable diseases. Nevertheless, vaccine adherence among pregnant women remains very low. The aim of our study was to evaluate the uptake of vaccines (influence, pertussis, and COVID-19) among women during pregnancy and to understand pregnant women’s knowledge of vaccines and the diseases they protect against. The purpose was to investigate the reasons why pregnant women chose not to be vaccinated and to develop effective strategies for informing them about the importance of vaccination for both maternal and fetal safety. *A* prospective observational study was conducted in the Department of Obstetrics and Gynaecology, “Ospedale Santa Maria della Misericordia” in Udine, from 1 December 2021 to 30 June 2022. During this period, a self-completed paper questionnaire was administered to women at the end of pregnancy or during the puerperium. A total of 161 questionnaires were collected. Higher educational level was found to be significantly associated with influenza vaccination uptake (*p* = 0.037, OR = 2.18, 95% CI 1.05–4.51). Similarly, for pertussis vaccination, adherence was mainly associated with higher educational level (*p* = 0.014, OR = 2.83, 95% CI 1.24–6.47), but also with Italian nationality (*p* = 0.003, OR = 3.36, 95% CI 1.56–8.43) and pregnancy attended by a midwife or private gynecologist (*p* = 0.028, OR = 0.39, 95% CI 0.17–0.90). Regarding the COVID-19 vaccine, the only factor positively influencing uptake was Italian nationality (*p* = 0.044, OR = 2.66, 95% CI 1.03–6.91). Women’s fear that vaccines would endanger the fetus appeared to be the most important reason for refusing vaccinations. Simultaneously, patients also exhibited a desire to receive more information about maternal vaccination, particularly from their general physician or gynecologist. For this reason, it is imperative to enhance maternal vaccination counselling, making it a routine step in prenatal care from the first antenatal visit until the postpartum period.

## 1. Introduction

Vaccinations play an essential role in antenatal care, safeguarding the health of the mother, the fetus, and the newborn against communicable diseases. Indeed, these three target groups are at higher risk of developing serious diseases that may lead to hospitalization or death. The preventive measure of vaccinating pregnant women has proven successful, as it not only prevents maternal morbidity and mortality but also reduces the risk of fetal illness and provides protection to the newborn during the initial months of life through passive immunity [1,2,3]. Although pregnancy is characterized by an immune-attenuated state with altered maternal immune responses to various antigens, studies have demonstrated that vaccines maintain comparable efficacy in pregnant women compared to nonpregnant women [4]. Given the safety and immunogenicity of most vaccines, maternal immunization stands as the most effective measure for reducing the incidence of certain infectious diseases and their complications in both the mother and the child [5]. The World Health Organization (WHO) recommends that pregnant women be vaccinated against several diseases, including seasonal influenza, tetanus, and pertussis [6,7,8]. Other vaccinations may be recommended in pregnant women under special circumstances and following a risk–benefit analysis (e.g., pneumococcal polysaccharide vaccine, meningococcal polysaccharide vaccine, and hepatitis A and B vaccines) [9]. In addition, some vaccines are contraindicated for administration during pregnancy but are recommended before conception, during family planning. These include the measles, mumps, and rubella vaccine (MMR) and the varicella vaccine (V alone or MMRV). Currently, in alignment with these recommendations, three vaccines are recommended for administration in pregnant women in Italy: influenza vaccine, tetanus–diphtheria–acellular pertussis vaccine (Tdap), and COVID-19 vaccine. Concerning influenza, pregnant women exhibit heightened susceptibility to influenza virus infection, potentially leading to adverse effects on pregnancy outcomes [10]. Indeed, the relative risk for pregnant patient to be admitted to a hospital due to this virus infection is increased compared with the general population [10]. The safety of influenza vaccination among pregnant women and its positive effect in reducing the risk of contracting such infection make it imperative to recommend this vaccination in every trimester [10]. Pertussis, also known as whooping cough, is a severe respiratory infection that may be very dangerous especially for neonates [11]. Currently, Tdap is recommended because a repeated cycle during pregnancy allows antibodies to pass through the placenta and reach the fetus, providing protection in the early months of life. The Tdap vaccination among pregnant women has been found to reduce the risk of hospitalization in infants < 2 months old [12]. All available safety data on Tdap during pregnancy are certainly reassuring [11]. Finally, the spread of the COVID-19 pandemic and the development of a vaccine against this virus promptly led scientific societies to evaluate whether this vaccine could also be administered during pregnancy without causing adverse effects. Despite emerging evidence indicating that SARS-CoV-2 infection does not result in more severe outcomes when contracted during pregnancy [13,14], considering the safety profile of the vaccine, it has been recommended for all pregnant women, especially in the presence of risk factors [14,15]. Despite the proven benefits of vaccinating pregnant women, uptake is still insufficient in many European countries, including Italy [16]. The optimal timing for vaccination during pregnancy depends on considerations of safety, the impact on the infant’s immune response to vaccination, and clinical effectiveness. Additionally, determining the target of protection (whether it is the mother or the infant) and identifying when maximum protection should be attained are two crucial factors to take into account when scheduling vaccine administration [17]. If neonatal immunity is the goal (e.g., pertussis vaccination), knowledge of the transfer of maternal antibodies should guide the timing, with the ideal time for vaccination being the early third trimester. In fact, the substantial transfer of immunoglobulin G (IgG) does not take place prior to 30 weeks of gestation and the maternal IgG levels peak approximately four weeks after immunization [18]. Administrating the vaccine during this period allows for the attainment of maximal maternal antibody levels and antibody transfer just prior to expected delivery. In contrast, if the vaccine’s purpose is to protect both the mother and the child, as in the case of influenza vaccination, it should be administered seasonally, irrespective of gestational age [17,19]. The Global Advisory Committee on Vaccine Safety has confirmed the safety of inactivated virus, bacterial, or toxoid vaccines administered during pregnancy with the overall benefits outweighing the risks. Data on adverse events associated with vaccination during pregnancy have been rarely reported in historical studies or in pregnant women vaccinated during mass vaccination campaigns [5]. Ensuring the safety of vaccines for both the mother and newborn is a priority in the decision-making process for maternal vaccination, thus emphasizing the importance of healthcare professionals’ counselling [17]. Continuous assessment and reporting of adverse events following vaccination during pregnancy remain crucial, especially for relatively newly introduced vaccines. At the same time, it is essential to assess baseline pregnancy outcomes in unvaccinated women. The ethical considerations surrounding vaccine trials and the strict adherence to the precautionary principle during pregnancy have historically led to the exclusion of pregnant women from most vaccine trials, resulting in a lack of evidence of vaccine safety and efficacy in this specific target population. However, with the increasing prioritization of maternal vaccination, ethical issues have been reconsidered, and recently a significant development has begun to allow the inclusion of pregnant women in vaccine trials [17]. Despite approximately 90% of European countries recommending the administration of influenza vaccines for pregnant women, in 2014–2015 vaccination coverage remained low, with half of the countries reporting coverage rates of less than 10% [17]. Numerous factors may contribute to the failure to take up recommended vaccinations [20,21], including reluctance among pregnant women or the role of healthcare providers. Obstetricians and gynecologists often serve as the primary physicians for women of childbearing age and are thus in a privileged position to assess vaccination status and offer appropriate vaccination counselling. Although lack of knowledge about vaccine safety during pregnancy and the need for counselling remain barriers to vaccine administration, concerns about vaccine safety have improved, allowing vaccination to be incorporated into obstetrician–gynecologist practice [22]. Regarding the main barriers that make patients insecure about vaccines, the most common issues relate to maternal or newborn safety. Other factors include concerns about vaccine effectiveness, underestimation of disease severity, belief in the unnecessary vaccination of healthy individuals, insufficient knowledge, and social and convenience-related issues [17,20,21]. Thus, it is evident that both patient and healthcare-provider barriers must be investigated, understood, and overcome to achieve higher vaccination coverage.

## 2. Materials and Methods

This was a prospective observational study conducted in the Department of Obstetrics and Gynaecology of the Academic Hospital of Udine (Italy) between 1 December 2021 and 30 June 2022. We submitted a self-completed paper questionnaire to women who were hospitalized at the end of their pregnancy or during the postpartum period. Women with a good comprehension of Italian and a minimum age of 18 years, who had given their written, informed consent to participate in the study, were included. Patients not meeting these requirements were excluded from the study. The sample size was calculated based on the primary objective of assessing adherence to vaccination during pregnancy, in particular to influenza vaccination. Considering approximately 800 parturients over the 7-month of interest (derived from birth data in our Clinic from December 2020 to June 2021) and assuming a 5% adherence rate to influenza vaccination, to obtain an estimate with a 95% confidence interval and a precision of 5%, the expected sample size was 67 questionnaires. The primary objective of this study was to assess vaccine uptake among women during pregnancy or the preconception period. Secondarily, the study aimed to provide information about pregnant women’s knowledge of vaccines administered during pregnancy or in the preconception period, as well as the diseases these vaccines protect against. Finally, we aimed to investigate the reasons behind women’s decision not to undergo vaccination during pregnancy, in order to develop effective strategies to inform pregnant women about the importance of vaccination during this critical period.

Given the current absence of an appropriate instrument in the literature for evaluating vaccine hesitancy and adherence among pregnant women, the questionnaire was drawn up based on the existing literature [20,21,23] and was subsequently reviewed by two experts in the field who worked in the Clinic. The questionnaire consisted of sociodemographic data and 15 questions. The sociodemographic data included biographical data (age, country of origin, educational level, and occupation), clinical and obstetric history, medications currently taken, and allergies. Then, the main part of the questionnaire included questions about vaccine adherence, vaccine hesitancy, and risk perception. The English-translated version of the full questionnaire is included in the Appendix A. Descriptive analysis was performed for all variables in the questionnaire. For the assessment of adherence to each vaccination, a logistic regression model was used with only those characteristics significant in the univariate analysis. A chi-square test was used to analyze risk perception between vaccinated and nonvaccinated individuals. A *p*-value < 0.05 was considered statistically significant. This study was reviewed and approved by the Institutional Review Board of the University of Udine (Udine, Italy), Prot IRB: 050/2022.

## 3. Results

A total of 161 questionnaires were collected for this study. The mean age of the participants was 33.9 ± 5.3 years [min–max, 19–47]; 91 out of 161 women (56.5%) were younger than 35 years. Most women were Italian (132/161, 81.9%), 55.5% (89/160) had a university degree or higher, and 60.2% (97/161) already had one or more children from previous pregnancies. In general, 43.0% (65/151) received care from a medium- or high-risk outpatient clinic, while 57.0% (86/151) received care from a midwife or private gynecologist. During pregnancy, 27.9% (45/161) of the women were vaccinated against influenza, 74.5% (120/161) against pertussis, and 70.1% (113/161) against COVID-19. Only 38 out of 161 women (23.6%) received all three recommended vaccines. In general, 14.9% (24/161) of the women had not received any vaccine during the current pregnancy. Of the sociodemographic factors examined (Table 1), only educational level was found to be significantly associated with influenza vaccination, with a higher educational level facilitating adherence (OR = 2.18). For pertussis vaccination, adherence was mainly associated with higher educational level (OR = 2.83), Italian nationality (OR = 3.36), and pregnancy managed by a midwife or private gynecologist (OR = 0.39). For the COVID-19 vaccine, the only factor that influenced uptake was Italian nationality (OR = 2.66).

### 3.1. Vaccine Adherence and Hesitancy

Overall, 24 of 161 women (14.9%) reported that they had not been vaccinated during pregnancy. Regarding the preconception period, 53.6% (82/153) of the women reported having been vaccinated against rubella in their lifetime, and the percentage of women who had received at least one of the three vaccinations (influenza, pertussis, or COVID-19) was higher (74/132, 56.1%) than of women who had not received any of these vaccinations (7/20, 35.0%). The reasons given by women for reduced vaccine adherence are shown in Table 2.

Specifically, 29.2 percent (7/24) cited logistical and personal problems, 45.8% (11/24) declined vaccination on account of the influence of others, and 58.3% (14/24) cited personal beliefs. The comparison between vaccinated and non-vaccinated individuals in terms of risk perception shows that the two groups had different levels of complacency for both influenza and pertussis vaccination. The detailed responses to the risk perception questions are shown in Table 3.

Most women knew that influenza in pregnancy can be dangerous for both the mother and the fetus (120/149, 80.5%), and the percentage was higher in vaccinated women (42/45, 93.3%) than in those who were not vaccinated (*p* < 0.01). There was also a positive association between the fear of the effects of influenza on fetal health and vaccination behavior (40/45, 88.9%, *p* < 0.01). In contrast, among unvaccinated women, the reason for vaccine hesitancy against influenza was the fear of side effects of the vaccine during pregnancy (42/107, 39.3%, *p* < 0.05). Similarly, the belief that pertussis can be very dangerous for the newborn (114/116, 97.4%, *p* < 0.01), that the infant can die in the first months of life if the mother is not vaccinated (84/115, 73.0%, *p* < 0.01), and that the infant can be protected by the mother’s vaccination (116/117, 99.1%, *p* < 0.01) were the factors significantly associated with women’s decision to be vaccinated against pertussis.

### 3.2. Risk Perception

In terms of general knowledge about vaccines, more than a quarter of participants (30/105, 28.7%) acknowledged having limited understanding of vaccines, with similar percentages observed among both vaccinated (25/86, 29.1%) and non-vaccinated women (2/5, 26.3%). A minority (10.5%, 9/86) of women believed that vaccines had not undergone sufficient research. Only one woman expressed the belief that vaccination was riskier than contracting the disease, while another woman did not perceive vaccination as necessary if others were vaccinated (both of these women had received at least one vaccine dose against influenza, pertussis, or COVID-19). Nonetheless, the majority of women expressed a desire for more information from their gynecologist (119/149, 80.0%) or their general physician (55/149, 36.9%), with the latter being particularly true for the unvaccinated group (9/21, 42.9%). Other sources of information included vaccine leaflets (48/149, 32.2%), while a few women expressed interest in alternative information channels, such as awareness campaigns and social media (10/149, 6.7%).

## 4. Discussion

Maternal vaccination is the most cost-effective strategy to protect the mother, fetus, and newborn from vaccine-preventable diseases during pregnancy. Vaccine hesitancy, identified in 2019 by WHO as one of the greatest threats to global health that threatens to undo progress in combating vaccine-preventable diseases [24,25], is considered one of the most important factors affecting vaccination coverage. Vaccine hesitancy has been identified as a complex and context-specific phenomenon that varies by time, place, and vaccine and is influenced by factors included in the 3C model (complacency, convenience, and confidence) [20], recently proposed in a modified 5C version (confidence, constraints, complacency, calculation, and collective responsibility) [21]. This study analyzed the adherence to the vaccinations recommended in Italy during pregnancy and in the preconception period, pregnant women’s knowledge about the vaccines and the infectious diseases against which the vaccines protect, and the reasons given for not vaccinating.

### 4.1. Vaccination Adherence

Despite the recommendations of the 2017–2019 National Vaccination Plan, vaccination coverage among pregnant women in Italy remains low. A survey conducted in three Italian cities revealed vaccination coverage rates of 6.5% for influenza and 4.8% for pertussis during the 2017–2018 influenza season [26]. Another study conducted during the 2018–2019 season in Italy reported coverage rates of 15% for influenza and 61% for pertussis, indicating an increase in coverage compared with the previous season. However, only one-third of women received both vaccines [27]. As for COVID-19, a study conducted in January 2021 found that only 28.2% of the sample were willing to be vaccinated [28]. In contrast to these results, our study found a high vaccination rate among women hospitalized in the Department of Obstetrics and Gynaecology of the Academic Hospital of Udine, especially for the vaccinations against pertussis (74.5%) and COVID-19 (70.1%). Nevertheless, in our case it was confirmed that the adherence to influenza vaccination was low, especially compared with other European countries [16], though still much higher than at the national level. Our study was conducted after the spread of the COVID-19 pandemic, which may have influenced pregnancy vaccination adherence. Indeed, it is possible that the experience of the pandemic and fear of SARS-CoV-2 infection have raised awareness among the general population regarding the importance of vaccines for protecting against infectious diseases. Consequently, this may have influenced the attitude of pregnant women toward maternal immunization [29]. However, the low percentage of women who had received all three recommended vaccines highlights the need to find strategies to further improve maternal immunization coverage. When considering vaccination against rubella in the preconception period, the low uptake may be due to women not knowing their vaccination status, as this vaccine is usually administered in childhood. Nevertheless, colleagues have reported a significant percentage of women of childbearing age being susceptible to rubella, underscoring the necessity for implementing new strategies to enhance vaccine coverage [30]. In previous studies, influenza and pertussis vaccination coverage during pregnancy have been shown to be influenced by sociodemographic factors. Specifically, women with at least one child, a high level of education, and Italian nationality are more likely to be vaccinated [26,31,32,33], and our results are consistent with these findings relating the educational level to influenza vaccination. Our finding that women with high levels of education are more likely to be vaccinated may be related to better communication with healthcare professionals and better access to and interpretation of information sources [31]. We also found that pertussis vaccination is more readily accepted when the pregnancy is managed by a midwife or private gynecologist. To our knowledge, this is the first instance of such data. Regarding COVID-19, the majority of studies have indicated that higher education, having children from previous pregnancies, and not belonging to an ethnic minority are associated with higher vaccination adherence. Additional social factors favoring maternal immunization against COVID-19 include the mother’s young age (<35 years), prior vaccination history, and employment status [34]. In our study, the only sociodemographic factor impacting compliance with COVID-19 vaccination was Italian nationality, likely attributable to challenges by non-Italian women in accessing and communicating with healthcare services.

### 4.2. Reasons for Vaccination Refusal

Women’s fear that vaccines would endanger the fetus appears to be the most important reason for refusing vaccinations, while concerns about vaccines being dangerous to themselves were less common. Secondarily, logistical issues, such as lack of time for vaccination, and being influenced by others (e.g., memories of relatives feeling bad after vaccination) were cited as additional reasons. Another factor contributing to vaccine refusal by women was the lack of the necessary information from health professionals. These findings are consistent with previous studies, which also identified the lack of recommendations by health professionals and concerns about vaccine safety, particularly with regard to infant health, as major reasons for the refusal [23,26,31,33,35]. In addition, the negative impact of advice from partners or relatives on vaccination adherence remains a significant challenge, as confirmed by the review of Wilson et al. [32]. Although our results show that awareness of vaccine effectiveness and safety, as well as the risk of not vaccinating during pregnancy, is good, a small but still significant number of women believe that influenza illness during pregnancy is less likely, as previously reported by Adeyanju et al. [25]. Nevertheless, the majority of women were aware of the dangers of influenza to both the mother and the fetus, confirming the findings reported in the literature [31]. However, it is concerning that a high number of women were unaware that influenza illness during pregnancy is associated with a high rate of hospitalization. Women expressed concerns about the effects that influenza and pertussis may have on the fetus, which is a critical factor in vaccination adherence, in agreement with the reports of Karafillakis et al. [36]. The awareness of influenza vaccination protection has been confirmed [26], whereas the same problem seems to be more complicated for pertussis. Although our women appeared to know that vaccination protects the newborn from pertussis in the first months of life, this understanding is not as clear in other contexts [26,27,37]. Consequently, these disparities in knowledge regarding various diseases and their associated risks are likely to influence the varying rates of vaccination against one infection or another. This was also evident in our study, as not all vaccinated women underwent all three vaccinations under examination. Presumably, a lesser fear of the consequences of one infection, a greater apprehension regarding others, and the proliferation of the COVID-19 pandemic may have influenced a woman’s decision to undergo one vaccination over another. While concern about the safety of the vaccine for the mother and fetus is one of the main factors contributing to vaccine hesitancy, our participants, similar to what was reported by colleagues, did not seem to express worry about this [31,38]. However, many expressed fear of potential side effects associated with the influenza vaccine, which subsequently influenced their willingness to vaccinate [32].

### 4.3. The Role of Health Professionals

Given that vaccines are the best and safest measure to prevent adverse outcomes associated with vaccine-preventable diseases, emphasizing the importance of maternal vaccination for maternal and child health is imperative. Many studies have shown that the lack of recommendations by health professionals and access to validated information about safety and effectiveness are major barriers to vaccination [26,33,35]. Despite the fact that the majority of women in our study expressed trust in vaccines, a main finding was their desire to receive more information about maternal vaccination, particularly from their general physician or gynecologist, a result consistent with previous reports by colleagues [31,33]. Although the Internet and social media are also mentioned as popular sources of information, the effectiveness of the different communication modalities is not the same, and receiving information from physicians has been proven to be associated with higher levels of knowledge and uptake [26,31,32,33,35]. Indeed, women who have not received counselling on this topic are more likely to perceive vaccination during pregnancy as risky for themselves and their children, leading to lower vaccination rates, as observed in out study. Therefore, it is essential for health professionals to educate pregnant women about the importance of vaccination, given their expertise in maternal and child health [31]. Despite physicians being the most trusted source of information by pregnant women, previous studies have shown that many of them are reluctant to provide advice on maternal vaccination [35]. Hence, it is crucial to enhance the education of health professionals about vaccination during pregnancy and improve their communication skills to properly inform pregnant women [39]. Obstetrician-gynecologists, from both private and public services, serve as the primary sources of information and support for pregnant women throughout the nine months of pregnancy and beyond. Moreover, they are responsible, as far as possible, for the health of both the mother and the fetus. As a result, they are supposed to regularly evaluate the vaccination status of their patients, address any concerns or inquiries, and recommend necessary vaccines during pregnancy. Their consistent counselling efforts may have the potential to significantly mitigate vaccine hesitancy.

### 4.4. Limitations of the Study

One limitation of our study was the small sample size, partly due to the non-compliance of pregnant women in completing the questionnaire and partly related to the brief duration allotted for survey distribution. Although extending the observation period might have yielded more significant results, we chose to focus our attention on the influenza vaccination campaign and the third dose of the COVID-19 vaccine campaign. Additionally, the self-completion of the survey may have resulted in inaccurate or incomplete responses due to the absence of guidance on questionnaire completion. Furthermore, the unavailability of national reports on vaccination coverage in pregnant women prevented us from comparing our results with national data.

## 5. Conclusions

Recognizing the significance of attaining high vaccination coverage among pregnant women, it is essential to comprehend and address the factors influencing adherence to vaccination services. Thus, we propose the integration of immunization counselling for pregnant women into routine antenatal visits. The counselling should primarily focus on the benefits of vaccination, particularly on the health of the child, as this plays a pivotal role in maternal acceptance of vaccination. Health professionals should bear in mind that pregnant women depend on them for clinical guidance and prioritize their child’s health, and recognize that their recommendations are critical to promoting vaccination adherence during pregnancy. Furthermore, clinicians should ensure that patients have comprehended the counselling before concluding, and remain available to address any further questions. Maternal immunization is a safe and effective strategy for providing infants with passive immunoprotection. However, immunization rates among pregnant women remain variable and suboptimal. We believe that vaccination adherence can be enhanced through the presence of informational materials such as posters or leaflets in hospital waiting rooms, making vaccination easily accessible (for instance, by administering it during routine visits), and continuous training of healthcare professionals to improve their knowledge and confidence.

## Figures and Tables

**Table 1 vaccines-12-00427-t001:** Univariate and multivariate analysis of influenza, pertussis, and COVID-19 vaccination by sociodemographic characteristics of respondents.

Pertussis Vaccine Uptake							
Sociodemographic Characteristic of Participant				Univariate Analysis		Multivariate Analysis	
Variable		Category		OR 95% CI	*p*-Value	OR 95% CI	*p*-Value
		Respondent	Vaccinated				
Age	<35	91	71	1	0.165	-	-
	≥35	70	59	1.69 0.81–3.53		-	
Nationality	Other	29	15	1	0.003	1	0.124
	Italian	132	105	3.36 1.56–8.43		2.21 0.81–6.04	
Educational level	High school qualification or lower	71	43	1	0.001	1	0.014
	University degree or higher	89	76	3.81 1.79–8.11		2.83 1.24–6.47	
Parity	No other children	64	46	1	0.530	-	-
	One or more children at home	97	74	1.26 0.61–2.58		-	
Health professional attending the current pregnancy	Midwife or private gynecologist	86	74	1	0.002	1	0.028
	High risk or general outpatient clinic	65	42	0.29 0.13–0.65		0.39 0.17–0.90	
**Influenza Vaccine Uptake**							
**Sociodemographic Characteristics of Participant**				**Univariate Analysis**		**Multivariate Analysis**	
**Variable**		**Category**		**OR** **95% CI**	***p*-Value**	**OR** **95% CI**	* **p-** * **Value**
		**Respondent**	**Vaccinated**				
Age	<35	91	31	1	0.225	-	-
	≥35	70	28	1.54 0.77–3.07		-	
Nationality	Other	29	14	1	0.070	-	-
	Italian	132	41	2.82 0.92–8.61		-	
Educational level	High school qualification or lower	71	14	1	0.037	-	-
	University degree or higher	89	31	2.18 1.05–4.51		-	
Parity	No other children	64	14	1	0.165	-	-
	One or more children	97	31	1.68 0.81–3.48		-	
Health professional attending the current pregnancy	Midwife or private gynecologist	86	28	1	0.421	-	-
	High risk or general outpatient clinic	65	17	0.75 0.37–1.52		-	
**COVID-19 vaccine uptake**							
**Sociodemographic Characteristics of Participant**				**Univariate Analysis**		**Multivariate Analysis**	
**Variable**		**Category**		**OR** **95% CI**	* **p-** * **Value**	**OR** **95% CI**	* **p-** * **Value**
		**Respondent**	**Vaccinated**				
Age	<35	91	61	1	0.320	-	-
	≥35	70	52	1.42 0.71–2.84		-	
Nationality	Other	29	4	1	0.006	1	0.044
	Italian	132	99	3.21 1.40–7.36		2.66 1.03–6.91	
Educational level	High school qualification or lower	71	43	1	0.021	1	0.226
	University degree or higher	89	69	2.25 1.13–4.47		1.58 0.75–3.34	
Parity	No other children	64	44	1	0.746	-	-
	One or more children at home	97	69	1.12 0.56–2.23		-	
Health professional attending the current pregnancy	Midwife or private gynecologist	86	68	1	0.010	1	0.085
	High risk or general outpatient clinic	65	39	0.39 0.19–0.80		0.513 0.24–1.10	

**Table 2 vaccines-12-00427-t002:** Reasons for not being vaccinated reported by participants who were not vaccinated during the current pregnancy (n = 24).

	Total Number (%; 95% CI)
**Logistic problems**	
I did not have time	4 (16.7%; 6.7–35.8)
I did not have enough information	2 (8.3%; 2.3–25.9)
There was no vaccine available	1 (4.2%; 0.7–20.2)
**Personal history**	
I was sick after being vaccinated in the past	-
I was sick after being vaccinated in the past during pregnancy	1 (4.2%; 0.7–20.2)
I have a chronic illness that is a contraindication to vaccination	-
I have a chronic illness because of which I am afraid to be vaccinated	3 (12.5%; 4.3–31.0)
A family member/friend of mine was sick after a vaccination	4 (16.7%; 6.7–35.8)
I am taking medication that constitutes a contraindication to vaccination	-
**Influence by other people**	
No health professional has given me the information I need to get vaccinated	4 (16.7%; 6.7–35.8)
My partner/husband has advised me not to get vaccinated during pregnancy	3 (12.5%; 4.3–31.0)
Friends/relatives have advised me not to get vaccinated during pregnancy	3 (12.5%; 4.3–31.0)
I have read on the internet/in newspapers that it is not recommended to get vaccinated during pregnancy	3 (12.5%; 4.3–31.0)
It was reported on TV that the vaccine is dangerous and that it is not recommended to get vaccinated during pregnancy	1 (4.2%; 0.7–20.2)
**Personal belief**	
I am afraid that the vaccine could be dangerous for my health	3 (12.5%; 4.3–31.0)
I am afraid that the vaccine could be dangerous for my baby’s health	13 (54.2%; 35.1–72.1)
I do not need the vaccine because I do not have contact with people who are at risk of getting ill	1 (4.2%; 0.7–20.2)

**Table 3 vaccines-12-00427-t003:** Risk perception about influenza and pertussis. Only affirmative answers are reported.

Influenza					
	Vaccinated for Flu (n. 45)		Not Vaccinated for Flu (n. 107)		*p*-Value
	N	%	N	%	
Is it less likely to get influenza during pregnancy?	0	0.0	12	11.2	
Can influenza in pregnancy be dangerous for both mother and fetus?	42	93.3	78	72.9	*p* < 0.05
Are women who get influenza during pregnancy at higher risk of being hospitalized?	34	75.6	64	59.8	
Is the vaccine dangerous for the fetus?	2	4.4	7	6.5	
Am I afraid of the side effects of the vaccine in pregnancy?	9	20.0	42	39.3	*p* < 0.05
Am I afraid of the effects of influenza on me if I contract it during pregnancy?	25	55.6	48	44.9	
Am I afraid of the effects of influenza on the fetus if I contract it during pregnancy?	40	88.9	76	71.0	*p* < 0.05
Is the vaccine unable to protect me from influenza?	3	6.7	22	20.6	
Will the vaccine protect me from contracting influenza?	39	86.7	89	83.2	
**Pertussis**					
	Vaccinated for pertussis (n. 116)		Not vaccinated for pertussis (n. 38)		
	N	%	N	%	
Is it dangerous to contract pertussis during pregnancy?	98	84.5	32	84.2	
	Vaccinated for pertussis (n. 117)		Not vaccinated for pertussis (n. 38)		
	N	%	N	%	
Can pertussis be very dangerous for the newborn?	114	97.4	31	81.6	*p* < 0.05
	Vaccinated for pertussis (n. 115)		Not vaccinated for pertussis (n. 37)		
	N	%	N	%	
Can the infant die in the first few months after birth if I have not been vaccinated against pertussis?	84	73.0	11	29.7	*p* < 0.05
	Vaccinated for pertussis (n. 117)		Not vaccinated for pertussis (n. 37)		
	N	%	N	%	
Can pertussis vaccine be dangerous for me?	9	7.7	5	13.5	
Can pertussis vaccine be dangerous for the fetus?	15	12.8	9	24.3	
Does pertussis vaccine protect my baby in the first few months after birth?	116	99.1	30	81.1	*p* < 0.05

## Data Availability

The data presented in this study are available on request from the corresponding author.

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
