# Peer review of "Vaccination: Adherence and Hesitancy among Pregnant Women for COVID-19, Pertussis, and Influenza Vaccines"

_vaccines, 2024, doi:10.3390/vaccines12040427_

Round 1
Reviewer 1 Report
Comments and Suggestions for Authors
The authors seek to determine vaccine uptake and knowledge regarding vaccines and the diseases they protect against amongst pregnant women admitted to a hospital in Italy. They want to determine the reasons why some women do not get vaccinated.
The literature review is relevant and covers all the aspects covered in the study. The methods are well described and the analysis relevant to the objectives.
The results are well presented and correctly interpreted. The discussion integrates the literature with the finding of this study. The authors comprehensively discuss the limitations of this study and make relevant and practical recommendations based of the findings.
Comments on the Quality of English LanguageThe quality of the English used is great. There are 2 instances where a more appropriate English word should be used - line 116 - adherence rather than adhesion, and line 151 reasons referred could be replace by reasons given.
Author Response
Thank you very much for providing us with your opinion on our study and manuscript. We have made the language modifications as suggested.
Reviewer 2 Report
Comments and Suggestions for Authors
authors performed a single centre prospective observational study to evaluate vaccine uptake in women during pregnancy or in preconception period and their knowledge of vaccines. The mean limitation, as they admitted, was too small sample size which could have been overcomed by a multicentric study or al longer period of observation, as reported. Some questions in the questionnaire seem not to consider other choices (the option "other" or "no answer" could have been reported among all the questions options).
Author Response
Thank you very much for reviewing the manuscript and for giving your opinion.
Reviewer 3 Report
Comments and Suggestions for Authors
1. There are many types of vaccines, and the title of the paper should specify the vaccines investigated in this study.
2. The research background of the paper needs to be further clarified. The preface of the paper should provide sufficient evidence to explain the necessity and safety of these three vaccines before and during pregnancy, especially the evidence of the necessity and safety of COVID-19 vaccine. The probability and evidence of the risk of infectious diseases arising from not receiving these three vaccines before and during pregnancy. What is the current international situation of pregnant women not receiving these three vaccines before and during pregnancy? What is the current situation in Canada where these three vaccines have not been administered domestically? What are the cases and proportions of pregnant women and infants in Canada who have not received these three vaccines developing infectious diseases?
3. How was the sample size determined for this study? Is the 161 participants in this study representative? From the research results, it appears that the sample size of this study is too small and the results are unreliable.
4. How is the quality control of the research survey questionnaire carried out? The paper needs to provide reliability and validity analysis of the survey results.
5. This study investigated the influencing factors of vaccine adherence and hesitancy among pregnant women before and during pregnancy (Tables 2 and 3), but the abstract of the paper only pointed out the education level and Canadian nationality (which is only general information of the surveyed population), without mentioning the factors that affect vaccine adherence and hesitancy. The conclusion section of the paper does not provide substantial conclusions.
6. The focus of this study should be on analyzing factors that affect vaccine compliance and hesitancy, rather than protective factors, with the aim of providing targeted policy recommendations.
7. Please check if the data in "Line146-152" is correct.
8. From the paper, it can be seen that apart from 24 cases who did not receive any vaccines, there is still a portion of pregnant women who have only received one or two vaccines (without fully receiving three vaccines). Why only one or two vaccines were administered instead of three, there are also many influencing factors that need to be considered. Suggest further comprehensive analysis.
Comments on the Quality of English LanguageModerate editing of English language required.
Author Response
Thank you for giving us your suggestions. We have revised the manuscript accordingly and hope that the changes have enhanced its quality.
- We changed the title
- As recommended, we have enhanced the introduction section to provide a comprehensive overview of vaccines, including details on their safety profile and the rationale behind their recommendation during pregnancy. We have limited ourselves a bit to avoid going off-topic.
- We specified how the sample size was determined.
- Considering that there are no perfect questionnaire in the literature we explained how was drown our questionnaire
- In the abstract, we solely highlighted the most significant factors identified in our analysis.
- The conclusions section has been expanded to include targeted policy recommendations.
- We checked and it is correct
- In the conclusions, we also addressed this fact, namely, that some patients received vaccination for only one or two of the three vaccines.
Round 2
Reviewer 3 Report
Comments and Suggestions for Authors
1.The author has responded well to the comments of the reviewers and made corresponding revisions to the paper.
2.Please check and verify the results in the abstract.(Line25-27):but also with Italian nationality and pregnancy attended by a midwife or private gynaecologist (p=0.028, OR=0.386, CI95% 0.165-0.903). This shows that there are two factors(Italian nationality and pregnancy attended by a midwife or private gynaecologist) related to the adherence for pertussis vaccination,but only one of OR and 95% CI were provided,and the OR and 95% CI show a negative correlation.
3.The main reasons for vaccination refusal need to be demonstrated in the results of the abstract. The abstract lacks conclusions.
4.The author should first elaborate on the conclusions drawn based on the results of this study in the conclusion section of the paper, and then propose countermeasures.
Comments on the Quality of English LanguageMinor editing of English language required.
Author Response
- Thank you very much
- We checked and corrected the results: now we have approximated all the results to the second decimal place exactly as they are reported in the table. Moreover, we added the missing OR and CI 95% for Italian nationality. The negative correlation with high risk or general outpatient clinic is correct (consequently positive with private gynaecologist or midwife).
- We improved the abstract
- Thank you for your comment. Except for the beginning of the discussion where we provided a brief introduction, the rest of our conversation focused on commenting on our data and referencing existing bibliographic sources.